# Three in One with Dual-Functional Hydrogel of Lactoferrin/NZ2114/LMSH Promoting *Staphylococcus aureus*-Infected Wound Healing

**DOI:** 10.3390/antibiotics13090889

**Published:** 2024-09-15

**Authors:** Kun Zhang, Xuanxuan Ma, Da Teng, Ruoyu Mao, Na Yang, Ya Hao, Jianhua Wang

**Affiliations:** 1Gene Engineering Laboratory, Feed Research Institute, Chinese Academy of Agricultural Sciences, Beijing 100081, China; 18797366561@163.com (K.Z.); 82101209114@caas.cn (X.M.); tengda@caas.cn (D.T.); maoruoyu@caas.cn (R.M.); nana_891230@126.com (N.Y.); haoya@caas.cn (Y.H.); 2Innovative Team of Antimicrobial Peptides and Alternatives to Antibiotics, Feed Research Institute, Chinese Academy of Agricultural Sciences, Beijing 100081, China; 3Key Laboratory of Feed Biotechnology, Ministry of Agriculture and Rural Affairs, Beijing 100081, China

**Keywords:** injectable hydrogel, Lf/NZ2114/LMSH (Three in One), would dressing, *Staphylococcus aureus* infection

## Abstract

Wound infections caused by *Staphylococcus aureus* often result in localized suppurative lesions that severely impede the healing process, so it is urgent to develop a dress with efficient antimicrobial and pro-healing functions. In this study, the bifunctional injectable hydrogel lactoferrin (Lf)/NZ2114/lithium magnesium silicate hydrogel (LMSH) was first successfully prepared through the electrostatic interaction method. The physical, biological, and efficacy properties are systematically analyzed with good shear-thinning capacity and biocompatibility. More importantly, it inhibits infection and promotes wound healing in a mouse wound infection model after 14 d treatment, and the bactericidal rate and healing rate were over 99.92% and nearly 100%, respectively. Meanwhile, the massive reduction of inflammatory cells, restoration of tissue structure, and angiogenesis in mice showed the anti-inflammatory and pro-healing properties of the hydrogel. The healed wounds showed thickening with more hair follicles and glands, suggesting that the hydrogel Lf/NZ2114/LMSH (Three in One) could be a better dressing candidate for the treatment of *S. aureus*-induced wound infections.

## 1. Introduction

Wound infection is a frequent disease in humans and animals (including pets). Usually, some purulent bacteria such as *Staphylococcus aureus* (*S. aureus*) invade the skin by mechanical, thermal, and other damages, causing local inflammation and suppuration, delaying wound healing, and even endangering the life and health of the individual or animal [1,2,3,4]. Therefore, an anti-infective and pro-healing bifunctional therapeutics was needed.

Currently, traditional antibiotics are usually used to kill the bacteria in a wound, but have significantly reduced therapeutic effect due to the emergence of resistant bacteria [5,6,7]. Some antibiotic alternatives combined with new materials have been studied such as silica, nanogold, and nanosilver. These materials have a better antimicrobial effect and penetration ability for drug-resistant bacteria, but their synthesis methods are complicated and their toxicity problems have not been resolved [7,8]. Wound dressings are considered as one of the most effective strategies to protect wounds from bacterial infection and moisture loss [9,10]. Among the wound dressings, hydrogels have received extensive attention from researchers for their better moisturization and biocompatibility properties, as well as their ability to act as a carrier for the delivery of body systems, which provide a moist environment that can promote wound healing [11,12]. However, traditional hydrogels exist in a specific form and need to be fixed to the wound location which may cause a longer healing cycle and discomfort. For this reason, it is important to prepare an effective antimicrobial and highly adaptable hydrogel dressing [13,14].

Injectable hydrogels are widely used for drugs, bioactive small molecules, and cellular delivery systems because of their ability to inject specific sites with low toxicity compared to the original hydrogels [15,16]. They are mainly formed by physical and chemical cross-linking, in which the former has the advantages of simple operation, no need for a catalyst, and no heat production compared to chemical cross-linking, to achieve the efficient absorption and utilization of the specific site [17,18,19]. They mainly include temperature-sensitive gels, pH gels, and electrostatic interaction gels, which are widely used as drug carriers and wound dressings for the treatment of wound infections, in which collagen (COL) and hyaluronic acid (HA) form hydrogels and intervene with gallic acid to have significant antioxidant functions and alleviate wound-induced oxidative damage [20,21,22]. Chitosan combined with glycerol phosphate or sodium alginate form an injectable hydrogel to promote wound angiogenesis and healing [3]. 

This paper is devoted to the preparation of a bifunctional (anti-inflammatory and antimicrobial) injectable hydrogel, while the hydrogel preparation methods are mainly classified into three types, among which the electrostatic interaction method is the simplest to prepare at a low cost. The Lf is mainly extracted from bovine milk and is active and expensive. It has gained attention as a candidate for wound treatment due to its multi-function as an anti-inflammatory, immunomodulatory, and antimicrobial agent. Although the Lf and LMSH matrix physically crosslinked can form an injectable hydrogel for the treatment of a wound by promoting wound angiogenesis, antimicrobial properties, and healing efficacy, the high cost limits its clinical application [23]. The antimicrobial peptide is a small peptide molecule, an important part of the body’s defense system, with cationic characteristics that can interact with the negative charge on the bacterial surface to quickly kill bacteria. This process does not easily induce drug resistance in bacteria, provides a next-generation alternative to antibiotics, and has received extensive attention from researchers [24,25]. NZ2114 is an excellent plectasin-derived peptide from *Pseudoplectania nigrella*, with a molecular weight of 4400 Da, PI of 8.4, and it behaves as a cationic peptide in environments below the isoelectric point [26]. It has a strong inhibitory effect on gram-positive bacteria, especially *S. aureus*, and has good biocompatibility at appropriate concentrations, which makes it a candidate for wound treatment. NZ2114 cannot form a hydrogel by itself and is unstable in vivo, so it can be used to form a hydrogel through the interaction between the positive charge and negative charge on its surface. The Lf is an iron-binding glycoprotein with a molecular weight of 70–80 kDa, an isoelectric point of 8.4–9.0, and a positive charge over a wide pH range [27]. The Lf is one of the multifunctional proteins in mammals, with anti-inflammatory, immunomodulatory, anticancer, antiviral, and antibacterial functions; the first two are the main functions of lactoferrin, also reported to have antibacterial activity, but requires a high mg/mL concentration level, increasing the cost of drugs and causing application difficulties. The use of Lf in combination with other drugs is a better choice [28,29,30,31]. The antimicrobial and anti-inflammatory properties of Lf have the potential to be a candidate for the treatment of wound infections, but low adsorption, weak stability, and short duration of efficacy when used alone requires carrier delivery such as hydrogels. Because of the cationic surface, it can be crosslinked with an anionic gel matrix to form an injectable hydrogel [32,33]. 

The LMSH is an excellent hydrogel matrix material with negatively charged sheet surfaces and positively charged edges, forming a hydrogel with a three-dimensional mesh structure through the unique mutual attraction of positive and negative charges [34]. It is also found to have antibacterial, anti-inflammatory, antitumor, antioxidant, and immunomodulatory functions. The LMSH also exhibits better biocompatibility, promotes cell adhesion and proliferation, promotes platelet agglutination, and exerts hemostatic effects [35,36,37]. The negative surface charge of LMSH can interact with the positive surface charge of lactoferrin and antimicrobial peptide NZ2114 to easily form an injectable hydrogel, which meets the requirements of antimicrobial and pro-healing as part of a high-quality wound dressing [38,39,40,41].

In this study, the bifunctional injectable hydrogel Lf/NZ2114/LMSH as a wound dressing including physicochemical properties (antimicrobial, anti-inflammatory, and biocompatible) was prepared using electrostatic interaction. The hydrogels can be used as lactoferrin and NZ2114 delivery carriers, and their good shear-thinning ability and porous mesh structure enable sustained drug release at the wound site to promote healing. With good antimicrobial, anti-inflammatory, biocompatible, and pro-angiogenic characteristics, this gel “Three in One” may be an excellent dressing candidate for promoting *Staphylococcus aureus* infection wound healing.

## 2. Results and Discussions

### 2.1. Preparation and Characterization of Lf/NZ2114/LMSH

#### 2.1.1. Preparation Scheme of Lf/NZ2114/LMSH

The 3D structures of Lf and the antimicrobial peptide NZ2114 have been well studied, where the lactoferrin sequence contains a 700 amino acid peptide chain consisting of two symmetrical globular regions at the N-terminal and C-terminal ends, each of which contains an iron-binding site with an isoelectric point of 8.7; its N-terminal end is a multifunctional region of Lf, with a large amount of positive charge distributed on the surface [42]. The sequence of the antimicrobial peptide NZ2114 contains a 40 amino acid peptide chain with a typical CSαβ structure, an isoelectric point of 8.4, and a positive charge of 3.5 distributed on its surface [43]. When LMSH is dissolved in water, the surface of the lamellae is negatively charged with a small positive charge at the edges due to the protonation of Mg-OH [44]. The positively charged Lf/NZ2114 was homogeneously mixed with the negatively charged LMSH solution, and the 1.5% Lf/NZ2114/LMSH and 1% Lf/NZ2114/LMSH hydrogels were prepared by electrostatic interaction at room temperature (Graph abstract and Figure 1). The physical and chemical properties were studied comprehensively and systematically, and their effects on wound infections were investigated for antimicrobial, anti-inflammatory and pro-healing qualities, aiming to provide an antimicrobial and pro-healing injectable hydrogel wound dressing.

#### 2.1.2. Formulas and Good Antimicrobial Activity of Lf/NZ2114/LMSH

At room temperature, Lf and NZ2114 were able to form a milky, pale-yellow solution with LMSH where 0.25% LMSH, 0.5% LMSH could not form a gel, and 1% LMSH and 1.5% LMSH were able to form a hydrogel within 5 s after a rapid sol–gel transition (Figure 1a). It was observed that the hydrogel preparation of 1% Lf/NZ2114/LMSH showed less effect on the antimicrobial activity of NZ2114; no significant effect of a 3% Lf/NZ2114/HACC on antimicrobial activity was observed by inhibition zone assay (Figure 1b). The above results show the successful preparation of hydrogels, which can efficiently inhibit *Staphylococcus aureus* with a slight reduction of antimicrobial activity related to possible partial neutralization of the positive charge of AMP and its slow release.

#### 2.1.3. FIR Analysis of Lf/NZ2114/LMSH (Three in One)

The synthetic hydrogels were characterized by FTIR analysis (Figure 1c). The LMSH displayed characteristic absorption peaks at 3366 nm and 1055 nm, si–o and -OH bonds, and chitosan showed characteristic absorption peaks at 1420 nm, 1550 nm, and 1650 nm for a methyl group and amide group I and II, respectively [45]. Meanwhile, Three in One Lf/NZ2114/LMSH showed characteristic peaks of Lf, NZ2114, and LMSH at the corresponding wavelength, indicating that the three materials were successfully compounded by using the simple physical means described in the previous section without chemical interaction.

#### 2.1.4. Lamellar Porous Mesh Structure of Lf/NZ2114/LMSH (Three in One)

Scanning electron microscopy showed that the surface morphology of 1% Lf/NZ2114/LMSH had a lamellar porous mesh structure (Figure 2), which acts as a good transport channel for nutrients or drug small molecules, and can be effectively delivered and released to play a role in promoting cell proliferation and wound healing [46]. In contrast, 0.5% Lf/NZ2114/LMSH had a large lamellar structure, indicating that the self-assembly of LMSH at a low concentration reduces the negative charge, which prevents the formation of hydrogel by attraction from electrostatic interactions. The specific surface area of the lamellar structure increases the contact area with bacteria, and its surface electrification also promotes the interaction with the surface charge of the bacterial membrane, which gives it certain anti-inflammatory and antibacterial activities. Meanwhile, 3% Lf/NZ2114/HACC presents large lamellar protruding structures, consistent with conventional encapsulation.

### 2.2. Better Shear-Thinning Characteristics of Lf/NZ2114/LMSH

A 1% Lf/NZ2114/LMSH can form hydrogel with typical injectable characteristics. The results of rheological characteristics show that with the increase in shear force from 0-4000 Mpa.s, 1% Lf/NZ2114/LMSH gradually and uniformly tends to decrease from 9–0 Pa, which has better shear-thinning ability, while 3% Lf/NZ2114/HACC does not show this characteristic due to its a poor gelation ability (Figure 3a,b). The results of rheological characterization of the 1% Lf/NZ2114/LMSH and 1.5% Lf/NZ2114/LMSH groups are shown in Figure 3c. The NZ2114/LMSH group shows storage modulus (G′) > loss modulus (G″) at the beginning, and the gels inclined to the elastic solid state, indicating that the hydrogel phase is more resilient [47]. Further, G′ decreases with shear running was in the range of 0.1–1, while G″ increases with running, and a sol–gel transition occurs at about 20% of shear stress, with severe disruption of the gel structure, transforming it from a viscoelastic solid to a liquid. Then, G′ and G″ are converted to a liquid after which G′ and G″ both showed a decreasing trend, further indicating that 1% Lf/NZ2114/LMSH has significant shear-thinning properties and can be injected into specific sites to exert effects. The 0.5% Lf/NZ2114/LMSH showed a similar trend, indicating that the preparation of LMSH, being homogeneous and consistent, did not affect the sol–gel transition behavior [48]. On the contrary, the 3% Lf/NZ2114/HACC did not have this property, and thus did not have an injectable gel property and could be used as a conventional gel [48] (Figure 3c).

### 2.3. Good Antimicrobial and Biocompatible Properties

One of the important properties of wound dressings is antimicrobial properties, which can prevent the bacterial infection of wounds and remove pathogenic bacteria from infected wounds to promote the wound healing process [49]. The Lf requires a high concentration to exert weak antimicrobial activity, and LMSH also has weak antimicrobial activity, which not only leads to a significant increase in the cost of the new product, but at the same time, the newly prepared gel will not exert superior antimicrobial properties [50]. Therefore, it is necessary to introduce the antimicrobial peptide NZ2114, which has better antimicrobial properties and biocompatibility, so that the prepared hydrogel has better antimicrobial properties [51]. The antimicrobial results of the prepared hydrogels are shown in (Figure 3d). The 1% Lf/NZ2114/LMSH had a stronger inhibitory effect on *S. aureus* CAAS-FRI-2023-02, with an inhibition rate of 99.99% close to that of NZ2114, indicating it can normally show activity after the formation of hydrogel, whereas the inhibition rate of *S. aureus* by using LMSH and/or Lf alone was about 99.99%, 42%, and 53%, respectively, which indicates that the introduction of the trace antimicrobial peptide NZ2114 significantly enhanced its inhibition on *S. aureus*. The preparation gel was lower than that of the antimicrobial peptide NZ2114 alone, and it is possible that part of the positive charge of the antimicrobial peptide NZ2114 was bound to LMSH, leading to a reduction in the charge acting with the bacteria and a slight decrease in the antimicrobial activity, but it can still completely eliminate *S. aureus* (inhibition rate of 99.99%). Generally, the experimental results showed that 1% Lf/NZ2114/LMSH has good antibacterial activity and can be used as a dressing for the treatment of wound infections.

The biocompatibility analysis of the injectable hydrogel is a prerequisite to ensure its safety for in vivo administration, and the hemolytic test and cytotoxicity test were performed, respectively [52]. The hemolytic test method refers to the previous work in this laboratory [41], and the results (Figure 4a,b) showed that the three key groups of treatment were almost non-hemolytic to mouse erythrocytes, and the rate of hemolysis was less than 5%, which showed better biocompatibility. The cytotoxicity of the prepared hydrogels was examined using MTT assay and Calcein–AM/PI double staining assay, respectively. On the first day, the cell survival rate of the 1% Lf/NZ2114/LMSH group was up to 95%, which was higher than that of Lf alone, NZ2114, and LMSH. Its lamellar mesh structure probably contributes to cell growth. On the second day, the changing trend of cell viability was similar to that of the first day, in which the cell viability of the 1% Lf/NZ2114/LMSH group, Lf + NZ2114 group, and Lf increased significantly. The cell viability was all higher than 100%, and the cell viability of the NZ2114 and LMSH groups alone had a decreasing trend, but was higher than 80%. On the fourth day, the change trend of each group was consistent with Day 1 and Day 2, which had better biocompatibility for human immortalized epidermal HACAT cells (Figure 4c). Further confirming the results of MTT assay, Calcein–AM/PI double staining assay for the live and dead cell determination was carried out. The results are shown in Figure 4d. The number of live cells increased with the incubation time in the control and treatment groups (in 1% Lf/NZ2114/LMSH group, Lf + NZ2114 group, LfNZ2114, LMSH group). The cell growth status was good, there was no obvious change in morphology, and only a small number of dead cells. It indicates that the prepared hydrogel has good biocompatibility. Next, an in vivo trial would be conducted to verify the treatment efficacy of the injectable hydrogel.

### 2.4. Antimicrobial and Pro-Healing Effects In Vivo

#### 2.4.1. Wound Infection Model and Treatment

To further evaluate the antimicrobial and healing effects of hydrogels on wounds, *Staphylococcus aureus*-infected Balb/c mouse wound models were set up. The demonstration treatment groups were 1% Lf/NZ2114/LMSH and 3% Lf/NZ2114/HACC, respectively, and blank control and model infection groups were set up. Skin homogenates were collected at 4 d post-treatment, diluted, and coated on MHA plates, and the bactericidal rate was over 99.92%. Meanwhile, wound images were recorded at 4 d, 8 d, 12 d, and 14 d after treatment, and the wound size was calculated by Image J.JS software (https://ij.imjoy.io/) for each group to calculate the wound healing rate. The results showed that the wound area changed significantly 4 d, 8 d, 12 d, and 14 d after treatment in each treatment group; there are significant difference in wound healing rate. The wound healing increased significantly with time at 4 d, 8 d and 12 d, but increased slightly from 12 d to 14 d in the 1% Lf/NZ2114/LMSH group, showing good treatment results around 12 d. The infecting wound showed extensive damage compared to the control group and inflammatory reaction before 8 d; the wound size was reduced by some extent after 8 d, the wound size was significantly reduced at 14 d, and the recovery of the wounds was slower in the blank (CK) and infection groups (positive). The healing rate of the Lf/NZ2114/LMSH group was significantly different from that of positive group at 4 d, 8 d, 12 d, and 14 d. Among all the groups, 1% Lf/NZ2114/LMSH had the best healing effect with a healing rate close to 100%, indicating that it exerted antibacterial and pro-healing action, and possesses the characteristics of a desirable wound dressing. The healing rates of the CK, positive, and 3% Lf/NZ2114/HACC groups after 14 d of treatment were 92%, 78% and 96%, respectively (Figure 5).

#### 2.4.2. HE Stain of Skin

Wound healing is a complex process that includes the following four processes: hemostasis, inflammatory response, proliferation, and tissue remodeling [53]. Wound infection hinders the normal progression of the four processes and seriously interferes with healing [5,54]. In addition to observing the wound images, HE staining was used to further reveal the wound healing process. Normal tissues had a clear and regular structure of the keratinized layer, granular layer, basal layer, dermis, adipose layer, and muscle layer, but for wounds and infections, there were obvious changes in the structure of the tissues, with the thickening of epithelial layer, the basal layer of cells changing from 2 to 4 layers, the necrosis of lymphatic vessels in the dermis, the blurring of the structure, and the appearance of redness in the blood vessels. Wounds and infected tissues were diffused with a large number of inflammatory cells, and severe inflammatory reactions resulted in the appearance of pestle-like structures. In some cases, lymphovascular necrosis was accompanied by a large number of inflammatory cells and lymphocyte infiltration [54]. After 4 d treatment of 1% Lf/NZ2114/LMSH, NZ2114 exerted excellent antimicrobial function, Lf exerted important anti-inflammatory function, the tissue inflammatory cells were reduced, and the inflammatory reaction was greatly reduced. On the 8th day after treatment, there were a large number of inflammatory cells in the wounds of the blank control group, and the tissue structure was irregular. After 1% Lf/NZ2114/LMSH treatment, the inflammatory cells were greatly reduced; the tissue structure was gradually clear and regular, thickened, with a large number of neovascularizations; and the glands were restored to normal status after 14 d of treatment (Figure 6).

### 2.5. The Pro-Healing Mechanism of Lf/NZ2114/LMSH

#### 2.5.1. The qPCR Analysis for Factors IL-6, CD31, EGFR, and VEGF of Skin

Neovascularization is a key element in tissue regeneration, providing nutrients and oxygen for cell and tissue growth and promoting cell proliferation and wound healing, in which the vascular endothelial cell marker protein (CD31), epidermal growth factor receptor (EGFR), and vascular endothelial growth factor (VEGF) are wound-healing-associated growth factors that play an important role in cell proliferation, angiogenesis, and epithelialization [55,56]. After 14 d of treatment with 1% Lf/NZ2114/LMSH hydrogel, the EGFR, CD31, and VEGF factors showed an increase in expression and neovascularization in tissues with increasing days of treatment. Angiogenesis can increase cellular proliferation and neo-tissue production, and promote wound healing (Figure 7).

#### 2.5.2. IHC Assay

Inflammatory factor expression in tissues is the body’s immune response to external stimuli and can respond to the inflammatory response after local trauma and infection [57]. Excessive inflammatory exudation, edema, and suppuration of tissues can cause excessive inflammatory response of the organism [58]. To further investigate the anti-inflammatory function of 1% Lf/NZ2114/LMSH hydrogel, the expression levels of inflammatory factors IL-6 were measured in the tissues, and it was found that the expression of inflammatory sub-levels significantly decreased in the treatment with the prepared hydrogel. A treatment of 1% Lf/NZ2114/LMSH can restore the normal inflammatory level of tissues and promote wound healing by inhibiting the expression of pro-inflammatory factors in tissues. Therefore, this Three in One can be used as a wound dressing to promote wound healing.

In conclusion, the injectable hydrogel Three in One of 1% Lf/NZ2114/LMSH has better shear-thinning, antibacterial, anti-inflammatory, and biocompatibility properties. In the mouse wound infection model, it exerted a better antibacterial, pro-angiogenic, and anti-inflammatory effect, thus greatly enhancing the wound healing rate close to 100% after 14 d of treatment, which possesses great potential for antibacterial and anti-inflammatory dual-energy dressings (Figure 8). Antibiotics are commonly used as antibacterial and anti-inflammatory materials, but facing bacterial resistance, new materials are needed. Lactoferrin is an anti-inflammatory and antibacterial multifunctional material, which is mainly anti-inflammatory and biocompatible. In addition, the antimicrobial peptide shows better antibacterial activity, low drug resistance and biocompatibility, and synergizes with the anti-inflammatory material lactoferrin, which can make the preparation of novel dressings with antibacterial and anti-inflammatory functions more feasible and allow for greater application potential. The injectable hydrogel dressing prepared in this study can be used for wound infection and can be precisely applied to various wound shapes while avoiding secondary damage to the wound.

## 3. Material and Method

### 3.1. Materials and Animals

The Lf and LMSH were purchased from Shanghai Yuanye Bio-Technology Co., Ltd. (Shanghai, China); NZ2114 (pdb number: 6K50) was prepared in our laboratory (>90% purity); chitosan quaternary ammonium salt and sodium alginate were purchased from Macklin Biochemical Co. Biochemical Co., Ltd., Shanghai, China; *S. aureus* CAAS-FRI-2023-02 were separated from Huanxian Sheep Farm Gansu, China; HaCaT cells were obtained from Peking Union Medical College (Beijing, China). The MTT (3-(4,5-dimethyldiazol-2-yl)-2,5-diphenyl tetrazolium bromide) was supplied by Sigma (Beijing, China). The calcein–AM/PI was purchased from Solepol (Beijing, China); other reagents were analytical grade.

The BALB/c mice (6–8 weeks, SPF) were purchased from the Vital River Laboratories (VRL, Beijing, China). Animal experiments strictly complied with the requirements for animal handling and welfare of the Laboratory Animal Ethical Committee and its Inspection of the Feed Research Institute of Chinese Academy of Agricultural Sciences (CAAS) (AEC-CAAS-20090609).

### 3.2. Method

#### 3.2.1. Preparation of Injectable Hydrogel Lf/NZ2114/LMSH

The Lf and NZ2114 were put in a 100 mL beaker, the appropriate ddH_2_O was added with final concentrations of 20 mg/mL, 40 mg/mL, 80 mg/mL, 160 mg/mL, 320 mg/mL and 200 µg/mL, 400 µg/mL, 800 µg/mL, and 1600 µg/mL, respectively, the solution was stirred in a magnetic agitator until fully dissolved, and the pH value of the solution was adjusted to 6–7. The LMSH was placed in a 50 mL beaker, the appropriate ddH_2_O was added, the solution was stirred in a magnetic agitator for 30 min, and completely dissolved by ultrasonication at room temperature for another 30 min. The Lf/NZ2114 was mixed with LMSH, and the final concentrations of LMSH were 0.5%, 1%, 1.5%, 0.5%, and 1%. The 1.5% lithium magnesium silicate (LMSH) was configured and mixed with final concentrations of 20 mg/mL, 40 mg/mL, 80 mg/mL, 160 mg/mL, and 320 mg/mL Lf, respectively, with thorough stirring and ultrasonication to homogeneity; the same process was used for NZ2114 with final concentrations of 200 µg/mL, 400 µg/mL, 800 µg/mL and 1600 µg/mL, respectively, with thorough stirring and sonication to desirable homogeneity. Different combinations of gel-forming states were observed, and the lowest gel-forming concentrations of LMSH, Lf, and NZ2114 were screened. Finally, it was found that 1.5%, 1% LMSH were ultrasonically mixed with 40 mg/mL Lf and 400 µg/mL NZ2114 to prepare Lf/NZ2114/LMSH hydrogels for follow-up studies. According to the previous laboratory report, 3% chitosan hydrogel can encapsulate NZ2114 to form a hydrogel. This experiment set the 3% HACC hydrogel encapsulated with the above concentration of Lf and NZ2114 to prepare hydrogel Lf/NZ2114/HACC as a control [3]. The obtained materials were further used for the following experiments.

#### 3.2.2. Antimicrobial Assay, FT–IR, and SEM of Lf/NZ2114/LMSH

Firstly, an inhibition zone assay was performed to initially screen the effect of hydrogels on the antimicrobial activity of Lf/NZ2114/LMSH, and the detailed manipulations were previously described [3]. Briefly, the prepared hydrogels (1% Lf/NZ2114/LMSH, 1.5% Lf/NZ2114/HACC) and controls (CK, Lf, NZ2114, Lf + NZ2114) were incubated with 30 µL of suspensions on MHA plates (containing logarithmic-phase *Staphylococcus aureus*, 1 × 10^6^ CFU/mL) for 16 h in a 37 °C incubator for observation of the inhibition zone.

The chemical structure of Lf, NZ2114, LMSH, HACC, Lf/NZ2114, 1.5% Lf/NZ2114/LMSH, 1% Lf/NZ2114/LMSH, and 3% Lf/NZ2114/HACC were detected by the BRUKER FT–IR TENSOR 27 (FT–IR, wavelength range 400–4000 cm^−1^) [23]. Briefly, all samples were dried and powdered with potassium bromide, ground and mixed, pressed, and placed into the instrument for examination). The surface morphology of the above samples was observed by scanning electron microscopy (SEM, Hitachi SU8000, Tokyo, Japan). Briefly, the lyophilized samples were immobilized on silicon wafers, sprayed with gold, and placed into the instrument for observation. All samples were freeze-dried.

#### 3.2.3. Rheological Analysis

The rheological properties including viscosity and shear-thinning of 1% Lf/NZ2114/LMSH, 1.5% Lf/NZ2114/LMSH, and 3% Lf/NZ2114/HACC hydrogels were observed by Physica Rotational Rheometer Model MCR 301 at room temperature [39]. Briefly, the hydrogels (2–3 mL) were taken and spread uniformly on the rheometer disc and a small amount of silicone oil was added to prevent water evaporation. The relationship of hydrogel between shear and viscosity was determined at different frequencies (0.01–100 s^−1^). The shear-thinning ability of hydrogel was also investigated at a fixed frequency (1 Hz) and shear-strain scanning range of low (0.1%)-high (1000%)-low (0.1%) to detect the storage modulus (G′) and loss modulus(G″) of samples in the group of 1% Lf/NZ2114/LMSH, 1.5% Lf/NZ2114/LMSH, and 3% Lf/NZ2114/HACC.

#### 3.2.4. Antimicrobial Properties

The antimicrobial activity (MIC values) of Lf, NZ2114, and LMSH on *S. aureus* CAAS-FRI-2023-02 was determined by a method previously described by CLSI 2021 guidelines [40].

Hydrogels were sterilized by cobalt 60 radiation and set aside. The Logarithmic stage bacteria was diluted to 1 × 10^6^ CFU/mL, and incubated with sterile hydrogel for 16–18 h. After that, a gradient dilution of individual samples was performed, and 100 µL of the samples were uniformly coated with the MHA plate, and then incubated at 37 °C for 16 h. The number of colonies was recorded in the incubator.

#### 3.2.5. Biocompatibility

The biocompatibility of the synthetic hydrogel was tested for hemolysis and cytotoxicity. The hemolytic assay has been described previously [41]. In short, fresh red blood cells were collected from normal mice, diluted to 8% concentration and co-incubated with aseptically prepared groundwater gels with PBS and 0.1% Trix-100 blank and positive group; all groups were set up in three replicates. The MTT assay was used to determine the cytotoxicity of gels to HaCaT cells with three replicates set up in each group, as has been described previously [3]. Optimal-state HACAT cells were diluted to 2 × 10^5^ cells/mL and co-incubated with the gel for 1 d, 2 d, and 4 d. After washing with PBS, Calcein–AM/PI live/dead cell double-staining kit was used to detect cell survival; 1 mL of cell volume was added with 1 µL of Calcein–AM and 3 µL of PI, and washed thrice with PBS, and observed by fluorescence microscopy (Excitation wavelength 490 nm, emission wavelength 545 nm).

#### 3.2.6. Animal Experiment of Wound Infection

Forty-eight female BALB/c mice, randomly divided into four groups, were sterilized and shaved before the experiment. The mice were anesthetized by isoflurane respiration and appropriate equal area sizes of whole skin were clipped. The uninfected group was the negative control, and the infected group had 100 µL of logarithmic *S. aureus* CAAS-FRI-2023-02 (2 × 10^8^ CFU/mL) uniformly applied to the wounds in the positive group. The treatment group was coated with 100 µL of 1% Lf/NZ2114/LMSH and 3% Lf/NZ2114/HACC hydrogel. Wound healing was observed and recorded at 4, 8, 12, and 14 days after treatment, while the wound area was analyzed and calculated using ImageJ software. The wound healing rate was calculated according to the following formula: healing rate = (A0 − At)/A0, (Note: “A0” is the initial wound area, “At” is the healed area.)

The skin tissues were collected at different times and fixed in 4% paraformaldehyde, HE staining was performed to observe the tissue sections, and the expression levels of inflammatory factors IL-6, vascular endothelial growth factor (VEGF), epidermal growth factor receptor (ECFR), and pro-angiogenic factor CD31 in the skin tissues were measured at the RNA level at the same time. The expression levels of inflammatory factors IL-6 and pro-angiogenic factor CD31 at the protein level were measured by immunohistochemistry assay.

#### 3.2.7. Statistical Analysis

GraphPad Prism (version 8, USA) was used to analyze the data, and *T*-test and ANOVA methods were used to determine statistical significance (* *p* < 0.05, ** *p* < 0.01, *** *p* < 0.001).

## 4. Conclusions

To prepare antibacterial, anti-inflammatory, and low-cost wound dressings, the injectable antibacterial/anti-inflammatory bifunctional hydrogel was prepared by successfully complexing trace antibacterial peptide NZ2114 and Lf with hydrogel LMSH Three in One through simple electrostatic interaction. The hydrogel has good injectable properties, antibacterial, anti-inflammatory, biocompatibility, and efficient pro-healing ability. Its wound healing rate is close to 100% and the wound is completely recovered after 14 d of treatment in a mouse wound infection model. Although the preparation of injectable hydrogels has great potential for application in the treatment of wound infections, further in-depth studies on the formulation and mechanism are needed at a later stage.

## Figures and Tables

**Figure 1 antibiotics-13-00889-f001:**
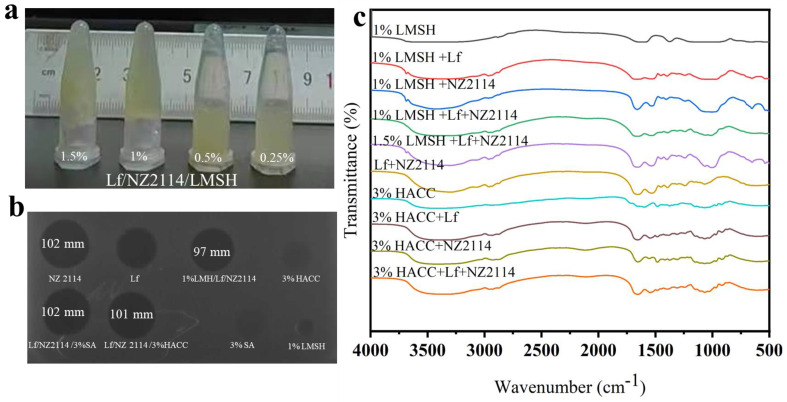
The photograph, antimicrobial activity and FT–IR spectrum of the injectable hydrogel Lf/NZ2114/LMSH. (**a**) The photograph of the synthetic injectable hydrogel; (**b**) Inhibition zone of NZ2114, Lf, 1% LMSH/Lf/NZ2114, 3% HACC/Lf/NZ2114, 3% SA, 1% LMSH; (**c**) The FT–IR spectra of 1% LMSH, 1% LMSH + Lf, 1% LMSH + NZ2114, 1% LMSH + Lf + NZ2114, 1.5% LMSH + Lf + NZ2114, 3% HACC, 3% HACC + Lf, 3% HACC + NZ2114, 3% HACC + Lf + NZ2114.

**Figure 2 antibiotics-13-00889-f002:**
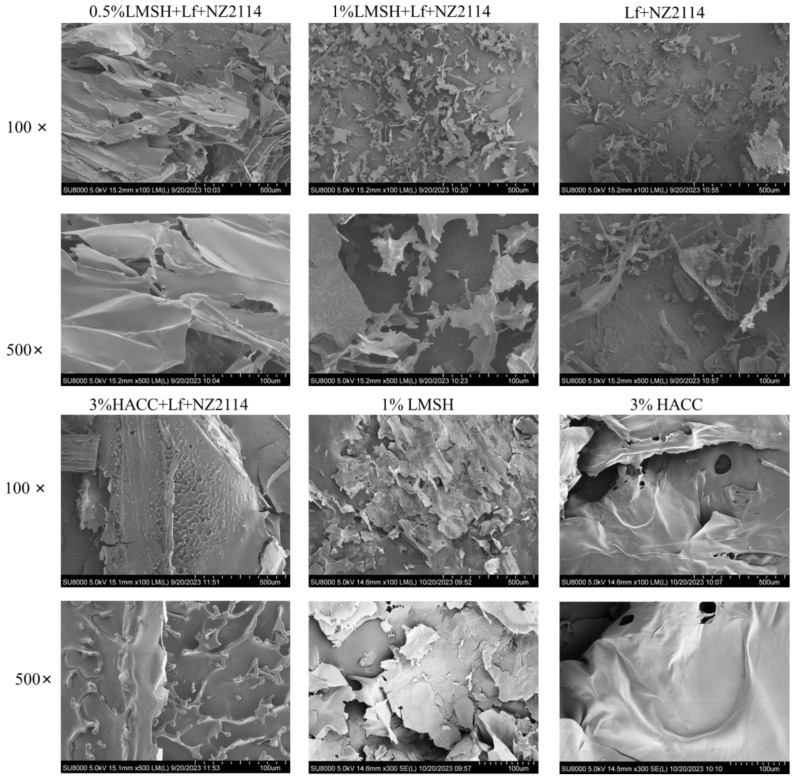
The morphology of the synthetic injectable hydrogel. The SEM image of 0.5% LMSH + NZ2114, 1% LMSH + Lf + NZ2114, 1% LMSH + Lf + NZ2114, 3% HACC + Lf + NZ2114, 1% LMSH, 3% HACC, Lf + NZ2114.

**Figure 3 antibiotics-13-00889-f003:**
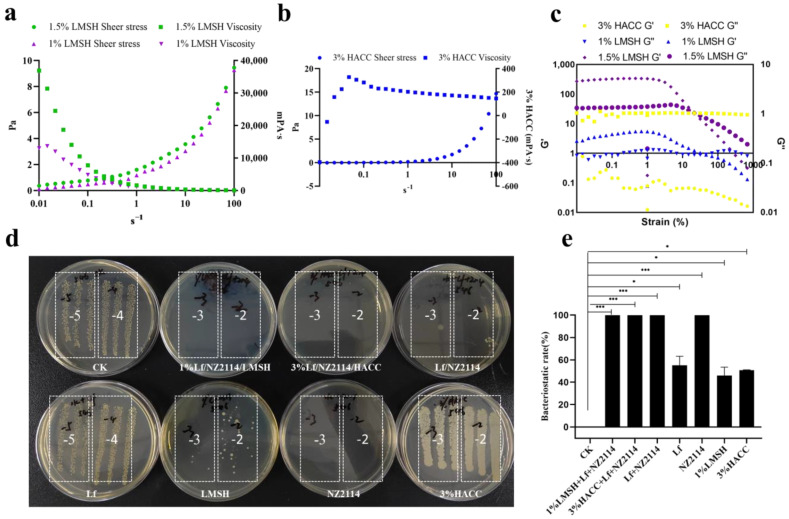
The viscosity, modulus, and bactericidal properties of different hydrogel samples. (**a**) The viscosity of 0.5% Lf/NZ2114/LMSH and 1% Lf/NZ2114/LMSH during sheer increase from 0.01 to 100 s^−1^; (**b**) The viscosity of 3% Lf/NZ2114/HACC during sheer increase from 0.01 to 100 s^−1^; (**c**) The storage modulus (G′) and loss modulus (G″) of 0.5% Lf/NZ2114/LMSH, 1% Lf/NZ2114/LMSH, and 3% Lf/NZ2114/HACC during strain increase from 0.1% to 1000% at the frequency of 1 Hz; (**d**,**e**) The bactericidal rate of 1% Lf/NZ2114/LMSH, 3% Lf/NZ2114/HACC, Lf + NZ2114, Lf, NZ2114, 1% LMSH, 3% HACC against *S. aureus* CVCC 546 (n = 3). These (−2, −3, −4, −5) are the number of dilutions. (* *p* < 0.05, *** *p* < 0.001).

**Figure 4 antibiotics-13-00889-f004:**
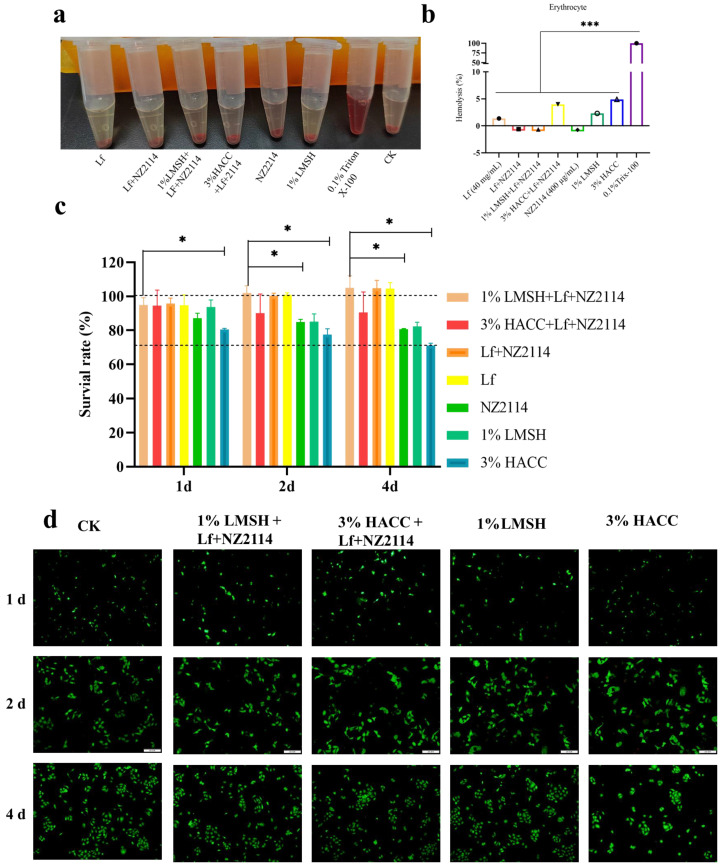
The biocompatibility of different hydrogel samples. (**a**,**b**) The images and hemolysis rate of Lf, Lf + NZ2114, 1% Lf/NZ2114/LMSH, 3% Lf/NZ2114/HACC, 1% LMSH, 3% HACC, 0.1% Trix-100; (**c**) Cytotoxicity of HACAT cells co-cultured with Lf, Lf + NZ2114, 1% Lf/NZ2114/LMSH, 3% Lf/NZ2114/HACC, 1% LMSH, 3% HACC. Samples at 1, 2, and 4 days, n = 3; (**d**) The images of calcein–AM/PI double staining of the HACAT cells that were incubated with 1% Lf/NZ2114/LMSH, 3% Lf/NZ2114/HACC, 1% LMSH, 3% HACC for 1, 2 and 4 days. (Scale bar = 100 μm). * *p* < 0.05, *** *p* < 0.001.

**Figure 5 antibiotics-13-00889-f005:**
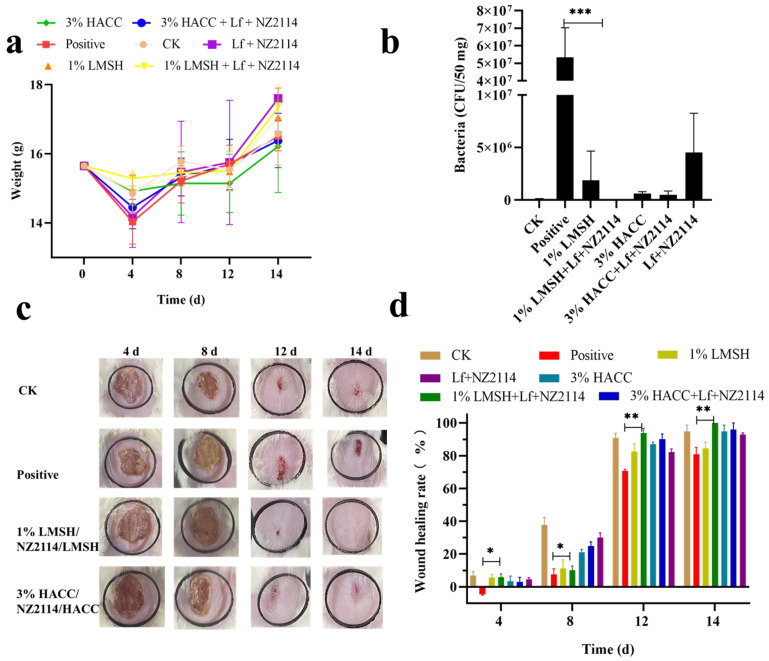
The body weight, skin load, and wound diagram of mice (n = 3). (**a**) The weight of mice untreated and treated with Lf + NZ2114, 1% Lf/NZ2114/LMSH, 3% Lf/NZ2114/HACC, 1% LMSH, 3% HACC for 0–14 days; (**b**) The bacteria of skin untreated and treated with Lf + NZ2114, 1% Lf/NZ2114/LMSH, 3% Lf/NZ2114/HACC, 1% LMSH, 3% HACC samples for 4 d; (**c**) The macroscopic images of wounds untreated and treated with 1% Lf/NZ2114/LMSH, 3% Lf/NZ2114/HACC samples for 4, 8, 12 and 14 d; (**d**) The wound healing rate of mice untreated and treated with Lf + NZ2114, 1% Lf/NZ2114/LMSH, 3% Lf/NZ2114/HACC, 1% LMSH, 3% HACC samples for 4, 8, 12, and 14 d. * *p* < 0.05, ** *p* < 0.01, *** *p* < 0.001.

**Figure 6 antibiotics-13-00889-f006:**
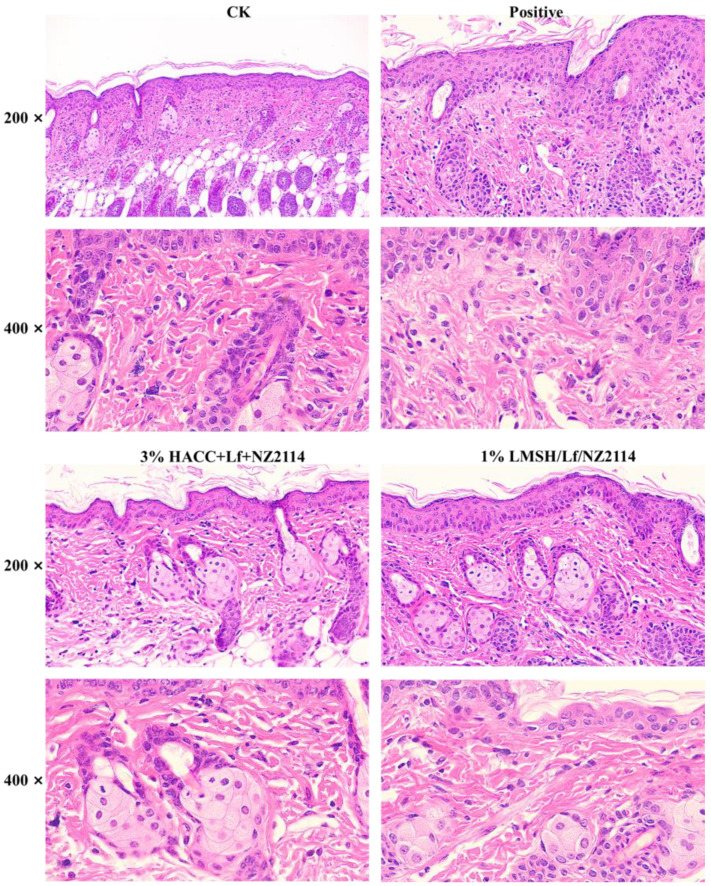
HE staining of the wounds without any treatment or treated with 1% Lf/NZ2114/LMSH and 3% Lf/NZ2114/HACC samples for 14 days.

**Figure 7 antibiotics-13-00889-f007:**
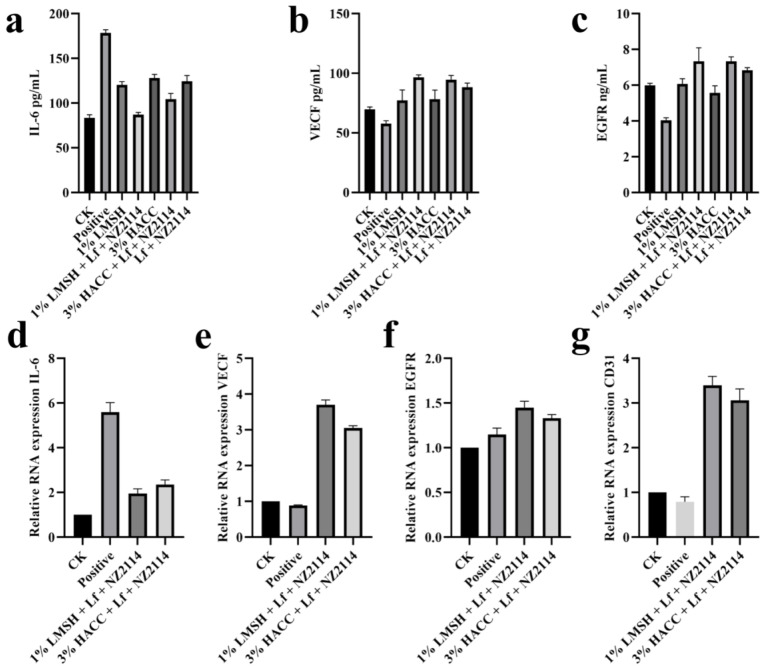
The cytokine secretion without any treatment or treated with 1% Lf/NZ2114/LMSH and 3% Lf/NZ2114/HACC samples for 14 days. (**a**–**c**) The expressions of IL-6, VEGF and EGFR were measured by ELISA kit. (**d**–**g**) The expressions of IL-6, VEGF, EGFR, and CD31 were measured at RNA level by qPCR.

**Figure 8 antibiotics-13-00889-f008:**
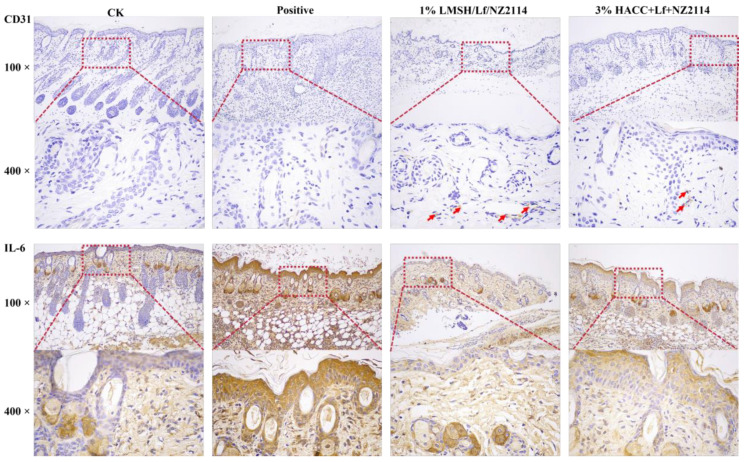
The immunohistochemistry of without any treatment or treated with 1% Lf/NZ2114/LMSH and 3% Lf/NZ2114/HACC samples for 14 days. CD31 staining of the wounds without any treatment or treated with different hydrogel samples for 14 days. IL-6 staining of the wounds for 14 days. (Red arrows are CD31 binding sites).

## Data Availability

All data generated or analyzed during this study are included in this published article.

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
