# Peer review of "Three in One with Dual-Functional Hydrogel of Lactoferrin/NZ2114/LMSH Promoting Staphylococcus aureus-Infected Wound Healing"

_antibiotics, 2024, doi:10.3390/antibiotics13090889_

Round 1
Reviewer 1 Report
Comments and Suggestions for Authors
The manuscript (antibiotics-3179797) submitted by Zhang et al. discussed about development of bifunctional injectable hydrogel exhibiting antimicrobial and wound healing properties. At many instances sentences are fragmented or unclear, likely due to typographical errors and careful proofreading and revision for grammar and sentence structure is required to enhance readability. Some of the specific comments are as follows:
1. Authors should consider breaking down the paragraphs in Introudction section into shorter and more focused paragraphs. For example, separate discussions on wound infections, current treatments, advantages of hydrogels, etc. can be added.
2. Before discussing specific materials like lithium magnesium silicate and the antimicrobial peptide, authors should provide a brief rationale for their selection.
3. The critical experimental conditions, such as the temperature and pH during the hydrogel formation should be clearly mentioned.
4. The potential reasons behind the variations observed in antimicrobial activity between hydrogel and their components should be discussed.
5. The impact of lamellar structure on the mechanical properties should be discussed in detail to give more clarity to the readers.
6. Authors should discuss the statistical significance of the wound healing results.
7. Phrase in the Line 367-368 “Different combinations of gel forming states were observed" is vague. The explanation on how observations were made should be included.
Comments on the Quality of English Language- Typographical errors and awkward phrasings should be avoided.
- Repetited phrases should be revised to improve clarity and eliminate redundancy.
Author Response
Reviewer 1
Comments and Suggestions for Authors
The manuscript (antibiotics-3179797) submitted by Zhang et al. discussed about development of bifunctional injectable hydrogel exhibiting antimicrobial and wound healing properties. At many instances sentences are fragmented or unclear, likely due to typographical errors and careful proofreading and revision for grammar and sentence structure is required to enhance readability. Some of the specific comments are as follows:
Question 1: Authors should consider breaking down the paragraphs in Introduction section into shorter and more focused paragraphs. For example, separate discussions on wound infections, current treatments, advantages of hydrogels, etc. can be added.
Answer 1: Thank you for your suggestion. I have broken down the paragraphs in Introduction section including five parts: wound infections, treatments, advantages of injectable hydrogel, selection of hydrogel components and brief conclusion.
Question 2: Before discussing specific materials like lithium magnesium silicate and the antimicrobial peptide, authors should provide a brief rationale for their selection.
Answer 2: Thank you for your suggestion. We have revised the introduction section and have given reasons before the specific materials discussion. The details are as follows:
“Lactoferrin (Lf) is an iron-binding glycoprotein with a molecular weight of 70-80 kDa, and a positive charge over a wide pH range[27]. Lf has gained attention as a can-didate for wound treatment due to its multi-function as an anti-inflammatory, immunomodulatory and antimicrobial agent. However, as an antimicrobial agent, Lf re-quires a high concentration of mg/mL to be effective, increasing the cost of drugs and application difficulties. The use of Lf in combination with other drugs is a better choice[28-31]. The antimicrobial peptides (AMPs) are small peptides molecule, as an important part of the body's defense system, with cationic characteristics that can in-teract with the negative charge on the bacterial surface to quickly kill bacteria, which is not easy to induce drug resistance in bacteria, and as a next-generation alternative to antibiotics has received extensive attention from researchers[24,25]. NZ2114 is an ex-cellent plectasin-derived peptide from pseudoplectania nigrella, with a molecular weight of 4400 Da, isoelectric point (pI) of 8.4, and behaves as a cationic peptide in environ-ments below the pI[26]. It has a strong inhibitory effect on gram-positive bacteria at low concentration, especially for S. aureus, and has good biocompatibility at appropriate concentrations. Therefore, the combined use of Lf and AMPs may provide better an-ti-inflammatory and antimicrobial functions for wound treatment. However, Lf and AMPs have low adsorption, weak stability and short duration of efficacy when used alone, requiring carrier delivery such as hydrogels. Due to the positive charge on the surface of the AMPs and Lf, they can be crosslinked with an anionic gel matrix to form injectable hydrogel [32,33]. Lithium magnesium silicate hydrogel (LMSH) is an excellent hydrogel matrix material with negatively charged sheet surfaces and positively charged edges, forming a hydrogel with a three-dimensional mesh structure through the unique mutual attraction of positive and negative charges[34]. It is also found to have antibac-terial, anti-inflammatory, antitumor, antioxidant, and immunomodulatory functions. The LMSH also exhibits better biocompatibility, promotes cell adhesion and val-ue-added, promotes platelet agglutination and exerts hemostatic effects[35-37]. Previ-ous study has shown that Lf and LMSH matrix physically crosslinked to form an in-jectable hydrogel for the treatment of wound, but the high cost limits its clinical appli-cation[23]. Therefore, reducing the amount of lactoferrin peptides by combining with antimicrobial peptides leads to cost reduction. The negative surface charge of LMSH can interact with the positive surface charge of Lf and AMP NZ2114 to form an injectable hydrogel, which meets the requirements of antimicrobial and pro-healing quality wound dressing[38-41].”
Question 3: The critical experimental conditions, such as the temperature and pH during the hydrogel formation should be clearly mentioned.
Answer 3: Thank you for your question. The pH and temperature were described at text. The pH value of the solution was adjusted to 6-7; The temperature was at room temperature.
Question 4: The potential reasons behind the variations observed in antimicrobial activity between hydrogel and their components should be discussed.
Answer 4: Thank you for your question. The reasons for the differences in antimicrobial activity of each hydrogel mainly include the following: in this paper, the antimicrobial peptide is used as the main antimicrobial material, lactoferrin and gel matrix (LMSH, chitosan) have weak antimicrobial properties, the positive charge on the surface of the antimicrobial peptide is able to bind to the negative charge on the surface of the bacteria, and when the antimicrobial peptide mixes with the LMSH, one of the positive charges is bound, and the antimicrobial activity will be slightly affected.
Question 5: The impact of lamellar structure on the mechanical properties should be discussed in detail to give more clarity to the readers.
Answer 5: Thank you for your question. The impact of lamellar structure on the mechanical properties was added in the manuscript. “The high specific surface area of the lamellar structure increases the contact area with bacteria, and its surface electrification also promotes the interaction with the surface charge of the bacterial membrane, which gives it certain anti-inflammatory and antibacterial activity. What's more, the hemostatic effect of LMSH should not be neglected because it can promote platelet coagulation and reduce the bleeding time.”
Question 6 Authors should discuss the statistical significance of the wound healing results.
Answer 6: Thank you for your suggestion. The statistical significance of the wound healing results
were discussed in the text. 1) “there are significant difference in wound healing rate. The wound healing increased obviously at 4 d, 8 d and 12 d, but increased slightly from 12 d to 14 d in 1% Lf/NZ2114/LMSH group, this show that the good treatment results around 12 d.” 2) “The healing rate of Lf/NZ2114/LMSH group was significantly different from that of positive group at 4 d, 8 d, 12 d, and 14 d.”
Question 7: Phrase in the Line 367-368 “Different combinations of gel forming states were observed" is vague. The explanation on how observations were made should be included.
Answer 7: Thank you for your question. The preparation standards are described in detail: Firstly, the mixture is observed by the naked eye to have a clear gelatinous state (colloidal, turbid), after which the container is placed upside down and the gel is not left behind, maintaining the original state unchanged.
Comments on the Quality of English Language
Question 1: Typographical errors and awkward phrasings should be avoided.
Answer 2: Thank you for your suggestion. Typographical errors and awkward phrasings were revised.
Question 2: Repetited phrases should be revised to improve clarity and eliminate redundancy.
Answer 2: Thank you for your suggestion. These repeated phrases were revised in this text.

Reviewer 2 Report
Comments and Suggestions for Authors
Review Report
In this work, the authors develop an injectable hydrogel based on the known formulation of lactoferrin and lithium magnesium silicate, with the incorporation of the peptide NZ2114, which is currently considered a promising alternative to antibiotics for treating infections. Lactoferrin was also found to be an excellent antimicrobial and antitumor agent, so the developed hydrogels are proposed as a dual delivery system for both lactoferrin and NZ2114. The hydrogels were obtained through electrostatic interactions and characterized employing FTIR, rheology, antimicrobial tests, etc. and in vivo antimicrobial and pro-healing determinations.
The antimicrobial results clearly demonstrate that the effect of NZ2114 is similar to that of the dual-functional hydrogel. However, further justification is needed to position the hydrogels as a real alternative to treat infected wounds. I recommend its publication in "Antibiotics" after a careful revision taking in consideration the following comments.
Major Comments:
1- The work presents interesting results and represents a significant insight into new hydrogels with antimicrobial properties for infected wounds against S. aureus. However, it would be beneficial to have a more detailed description of the formulations explored by the authors, as well as a clearer explanation of the rationale behind the choice of hydrogels for testing. It would be beneficial to have complete clarifications in this regard.
2- It would be helpful to have a clearer understanding of the compositions of the hydrogels that were selected, as there seems to be some confusion about which ones were chosen from all combinations dscribed in the experimental section. It would be helpful to have a more consistent approach to naming the compositions of the hydrogels and to provide more details about the proportions of the different components. It would be beneficial to establish a clear and univocal nomenclature for the materials in order to facilitate understanding and interpretation of the results. I wonder if it might be helpful to include a table for this purpose.
3- The presentation of results and discussions could be improved in terms of clarity and organization. I wonder if it might be helpful to consider providing clearer headings, subheadings, and transitions between sections. This could potentially enhance the readability and flow of the manuscript, making it easier for readers to follow the scientific rationale and conclusions.
4- The manuscript includes a great deal of data and analysis, which can sometimes make it seem a bit scattered and difficult to follow. It would be really helpful if the authors could find a way to link each section more closely with the others, especially in terms of how they present the data and the analysis.
5- It looks like many of the tests and results were based on just one repetition. Could the authors please explain how they obtained the standard deviation? This would help to provide more context for the experimental section.
6- It would be beneficial to review the captions to identify any potential mistakes or omissions. Figure 1-a. Could you kindly describe the differences between the hydrogel shown in the Eppendorf picture? Figure 1-c/b. Could you please elaborate on the importance and significance of FTIR analysis? It would also be greatly appreciated if you could identify the main characteristic bands in the figure. Figure 1-b/c. It would be beneficial to provide a detailed explanation of the results of the inhibition zone assay, and it may also be helpful to conduct control experiments with an antibiotic for comparison. It would be helpful to have a description of all components in the inhibition zones in the caption.
7- It would be greatly appreciated if all captions could be carefully revised, completed, and corrected.
8- It might be helpful to summarize the results of the diameter of inhibition zones in a table for easier understanding. It appears that the control hydrogel with chitosan and NZ2114 may have the most inhibitory effect. It would be beneficial for the authors to provide a more in-depth discussion and explanation of the results.
9- Section 2.2 – Lines 169-171 could perhaps be rephrased to better convey the intended meaning. It appears that the table referenced in this sentence was not included in the manuscript. At this point, it would be beneficial to include a supplementary section with additional information to help clarify any remaining questions for the readers.
10- Section 2.3 - It appears that, in contrast to the results shown in Figure 3.e, the Lf+LMSH+NZ2114 hydrogels may have somewhat reduced antimicrobial efficacy compared to NZ2114. It would be beneficial to the manuscript if the advantages of using hydrogels instead of peptides alone were clearly explained, with a focus on the benefits of the developed materials. The results of the in vitro experiment appear to contradict the assertion made in point 3 of the Highlight document.
11- Section 2.4. In light of the above observations, it may be beneficial to consider conducting in vivo experiments with NZ2114. This could help to shed further light on the effectiveness of Lf+LMSH+NZ2114 hydrogels as antimicrobials and wound healing-promoting materials, and potentially provide insights that could help to explain the differences observed in the in vitro results.
Minor comments:
- Acronyms must be clearly defined and should not be used in abstracts. Once an acronym is defined, it should be used instead of the complete word. The text contains inconsistencies. For example: Lactoferrin (Lf) is defined in line 62 on page 2, yet it is used in full on pages 74, 86, and 87, among others.
- The acronym LMSH refers to a "lithium magnesium silicate solution." It was defined in section 3, making it difficult for the reader to interpret. It should be defined once and at the beginning of the text.
- The text failed to define CK.
- Figure 2ddH2O+Lf+NZ2114 is not defined in the text.
- The text requires careful revision to improve the organization of ideas and address grammatical issues.
- There are instances when the caption and lettering in the figure do not align. For instance, Figure 1b and its caption correspond with the description label as c.
Comments on the Quality of English Language- The text requires careful revision to improve the organization of ideas and address grammatical issues.
Author Response
Reviewer 2
Review Report
In this work, the authors develop an injectable hydrogel based on the known formulation of lactoferrin and lithium magnesium silicate, with the incorporation of the peptide NZ2114, which is currently considered a promising alternative to antibiotics for treating infections. Lactoferrin was also found to be an excellent antimicrobial and antitumor agent, so the developed hydrogels are proposed as a dual delivery system for both lactoferrin and NZ2114. The hydrogels were obtained through electrostatic interactions and characterized employing FTIR, rheology, antimicrobial tests, etc. and in vivo antimicrobial and pro-healing determinations.
The antimicrobial results clearly demonstrate that the effect of NZ2114 is similar to that of the dual-functional hydrogel. However, further justification is needed to position the hydrogels as a real alternative to treat infected wounds. I recommend its publication in "Antibiotics" after a careful revision taking in consideration the following comments.
Major Comments:
Question 1: The work presents interesting results and represents a significant insight into new hydrogels with antimicrobial properties for infected wounds against S. aureus. However, it would be beneficial to have a more detailed description of the formulations explored by the authors, as well as a clearer explanation of the rationale behind the choice of hydrogels for testing. It would be beneficial to have complete clarifications in this regard.
Answer 1: Thank you for your suggestion. In this paper, 1% Lf/NZ2114/LMSH was finally identified as the preparative ground injectable hydrogel with the following formulation: 1% LMSH, 40 mg/mL lactoferrin, 400 µg/mL NZ2114. The purpose of this paper was to prepare injectable hydrogels by screening different formulations, determining their antimicrobial properties, structural appearance, rheological properties and in vivo effect tests to obtain good antimicrobial, sheet-forming pore-like structure, shear thinning and pro-healing hydrogels.
Question 2: It would be helpful to have a clearer understanding of the compositions of the hydrogels that were selected, as there seems to be some confusion about which ones were chosen from all combinations described in the experimental section. It would be helpful to have a more consistent approach to naming the compositions of the hydrogels and to provide more details about the proportions of the different components. It would be beneficial to establish a clear and univocal nomenclature for the materials in order to facilitate understanding and interpretation of the results. I wonder if it might be helpful to include a table for this purpose.
Answer 2: Thank you for your suggestion. The proportions of hydrogel were described in detail: “Lf and NZ2114 were put in a 100 mL beaker, The appropriate ddH2O was added with final concentration were 20 mg/mL, 40 mg/mL, 80 mg/mL, 160 mg/mL, 320 mg/mL and 200 µg/mL, 400 µg/mL, 800 µg /mL1600 µg/mL respectively, stirred in a magnetic ag-itator until fully dissolved, and the pH value of the solution was adjusted to 6–7. LMSH was placed in a 50 mL beaker, The appropriate ddH2O was added, stirred in a magnetic agitator for 30 min, and completely dissolved by ultrasonication at room temperature for another 30 min. The Lf/NZ2114 was mixed with LMSH, and the final concentration of LMSH was 0.5%, 1%, 1.5%. 0.5%, 1%, 1.5% lithium magnesium silicate (LMSH) was con-figured and mixed with final concentrations of 20 mg/mL, 40 mg/mL, 80 mg/mL, 160 mg/mL, 320 mg/mL lactoferrin (Lf) respectively by thorough stirring and ultrasonication to homogeneity; the same as, with final concentrations of 200 µg/mL, 400 µg/mL, 800 µg /mL1600 µg/mL NZ2114 were thoroughly stirred and sonicated to homogeneity. Different combinations of gel-forming states were observed, and the lowest gel-forming concentra-tions of lithium magnesium silicate, lactoferrin, and NZ2114 were screened. Finally, it was found that 1.5%, 1% LMSH were ultrasonically mixed with 40 mg/mL Lf and 400 µg/mL NZ2114 to prepare Lf/NZ2114/LMSH hydrogels for follow-up studies.” “The obtained materials were further used for the following experiments.”
Question 3: The presentation of results and discussions could be improved in terms of clarity and organization. I wonder if it might be helpful to consider providing clearer headings, subheadings, and transitions between sections. This could potentially enhance the readability and flow of the manuscript, making it easier for readers to follow the scientific rationale and conclusions.
Answer 3: Thank you for your suggestion. The results and discussions were improved in terms of clarity and organization. We added subheadings:
2.1. Preparation and characterization of Lf/NZ2114/LMSH
2.1.1 Preparation scheme of Lf/NZ2114/LMSH
2.1.2 Formulas and good antimicrobial activity of Lf/NZ2114/LMSH
2.1.3 FIR analysis of Lf/NZ2114/LMSH
2.1.4 Lamellar porous mesh structure of Lf/NZ2114/LMSH
2.2. Better shear thinning characteristics of Lf/NZ2114/LMSH
2.3. Good antimicrobial and biocompatible properties
2.4. Antimicrobial and pro-healing effects in vivo
2.4.1 Wound infection model and treatment
2.4.2 HE stain of skin
2.5. The pro-healing mechanism ofLf/NZ2114/LMSH
2.5.1 The qPCR analysis for factors Il-6, CD31, EGFR and VEGF of skin
2.5.2 IHC assay
Further, the transitions paragraph was added. “Next, an in vivo trial was conducted to verify the treatment efficacy of the injectable hydrogel in 2.3”.
Question 4: The manuscript includes a great deal of data and analysis, which can sometimes make it seem a bit scattered and difficult to follow. It would be really helpful if the authors could find a way to link each section more closely with the others, especially in terms of how they present the data and the analysis.
Answer 4: Thank you for your suggestion. The data and analysis in the paper focus on the formulation of the injectable hydrogel preparation, in vitro antimicrobial activity, biocompatibility (haemolysis, cytotoxicity), and in vivo efficacy (wound healing rate), and the sections are independent but interconnected, and we further increase the linkage between the data.
Question 5: It looks like many of the tests and results were based on just one repetition. Could the authors please explain how they obtained the standard deviation? This would help to provide more context for the experimental section.
Answer 5: Thank you for your question. Many of the replicated tests may not have been shown in the article or in the results: including antimicrobial activity and survival were set up with three replicates each in vitro; body weight, antimicrobial activity, and wound healing rates also were set up with three replicates for each group in vivo.
Question 6: It would be beneficial to review the captions to identify any potential mistakes or omissions. Figure 1-a. Could you kindly describe the differences between the hydrogel shown in the Eppendorf picture? Figure 1-c/b. Could you please elaborate on the importance and significance of FTIR analysis? It would also be greatly appreciated if you could identify the main characteristic bands in the figure. Figure 1-b/c. It would be beneficial to provide a detailed explanation of the results of the inhibition zone assay, and it may also be helpful to conduct control experiments with an antibiotic for comparison. It would be helpful to have a description of all components in the inhibition zones in the caption.
Answer 6: Thank you for your question. The captions have been revised. The Figure 1-a has been described in detail: “At room temperature Lf and NZ2114 were able to form a milky pale yellow solution with LMSH where 0.25% LMSH, 0.5% LMSH could not form a gel, and 1% LMSH and 1.5% LMSH were able to form a hydrogel within 5 s after a rapid sol-gel transition.”
This FTIR analysis can prove that the components of the injectable hydrogel are physically cross-linked and the characteristic chemical groups are not changed.
Fig 1b: “It was found that gel matrix (LMSH, HACC and SA) have weak antimicrobial properties (almost no anti-bacterial transparent ring), and the 1 % Lf/NZ2114/LMSH hydrogel displayed an anti-bacterial transparent ring, which had less effect on the antimicrobial activity of NZ2114, and the control 3% Lf/NZ2114/HACC antimicrobial activity was also found to have less effect by inhibition zone assay”. In this study, 3% Lf/NZ2114/HACC was the control.
Question 7: It would be greatly appreciated if all captions could be carefully revised, completed, and corrected.
Answer 7: Thank you for your suggestion. All captions were revised, completed, and corrected.
Question 8: It might be helpful to summarize the results of the diameter of inhibition zones in a table for easier understanding. It appears that the control hydrogel with chitosan and NZ2114 may have the most inhibitory effect. It would be beneficial for the authors to provide a more in-depth discussion and explanation of the results.
Answer 8: Thank you for your question. The diameter of inhibition zones were not determined by measurement, this experiment through the naked eye to observe the inhibition zones with or without obvious changes, if the formation of hydrogel antibacterial activity is affected by the change is obvious, it will be rounded off, do not need to be measured accurately. The antimicrobial peptides NZ2114, 1% Lf/NZ2114/LMSH were found to have diameters of 102 mm and 97 mm, respectively, which indicated that the antimicrobial activity was basically unaffected and could be used for subsequent experiments. The antimicrobial activity of Chitosan combined with NZ2114 to form hydrogel is greater than lithium magnesium silicate, but combined with rheological analysis found that chitosan combined with NZ2114 did not form a hydrogel, tends to be in the liquid state, so selected lithium magnesium silicate, the formation of injectable hydrogel has the function of slow-release long-lasting.
Question 9: Section 2.2 – Lines 169-171 could perhaps be rephrased to better convey the intended meaning. It appears that the table referenced in this sentence was not included in the manuscript. At this point, it would be beneficial to include a supplementary section with additional information to help clarify any remaining questions for the readers.
Answer 9: Thank you for your question. The lines 169-171 were rephrased, we made a writing mistake, the table was changed to Fig. 3c.
Question 10: Section 2.3 - It appears that, in contrast to the results shown in Figure 3.e, the Lf+LMSH+NZ2114 hydrogels may have somewhat reduced antimicrobial efficacy compared to NZ2114. It would be beneficial to the manuscript if the advantages of using hydrogels instead of peptides alone were clearly explained, with a focus on the benefits of the developed materials. The results of the in vitro experiment appear to contradict the assertion made in point 3 of the Highlight document.
Answer 10: Thank you for your question. Antimicrobial peptide NZ2114 in aqueous solution can be directly combined with bacteria to quickly kill bactericidal, but it is unstable in vivo; when combined with the LMSH, the antimicrobial peptide NZ2114 part of the charge and magnesium lithium silicate combination, part of the antimicrobial activity is shielded, but the formation of the injectable hydrogel is able to deliver the NZ2114 to a specific site to achieve the effect of slow-release and long-lasting effect. Therefore, the weak antimicrobial activity in vitro does not indicate a poor effect in vivo, this result is illustrated by Figure 5b, 1% Lf/NZ2114/LMSH hydrogel has better bactericidal effect in vivo.
Question 11: Section 2.4. In light of the above observations, it may be beneficial to consider conducting in vivo experiments with NZ2114. This could help to shed further light on the effectiveness of Lf+LMSH+NZ2114 hydrogels as antimicrobials and wound healing-promoting materials, and potentially provide insights that could help to explain the differences observed in the in vitro results.
Answer 11: Thank you for your question. The aim of this paper is to prepare a bifunctional injectable hydrogel and further in vivo validation of the therapeutic efficacy after in vitro validation of its characteristics, without setting up a separate NZ2114 group, further NZ2114 and Lf should be viewed as composite. The results of the former experiments in this paper verified that the NZ2114 was mixed with gel matrix to successfully prepare injectable hydrogel 1% Lf/NZ2114/LMSH, the antimicrobial activity was only slightly affected but the main antimicrobial activity was unaffected, so the prepared injectable hydrogel could be used for in vivo tests and there was no need to increase the NZ2114 group, and at the same time, the NZ2114 was effective against S.aureus induced wound infection. The effect was not as effective as the therapeutic effect after formation of the hydrogel and the article has been cited in the text.
Minor comments:
Question 1: Acronyms must be clearly defined and should not be used in abstracts. Once an acronym is defined, it should be used instead of the complete word. The text contains inconsistencies. For example: Lactoferrin (Lf) is defined in line 62 on page 2, yet it is used in full on pages 74, 86, and 87, among others.
Answer 1: Thank you for your question. All questions about acronyms were revised in this paper.
Question 2: The acronym LMSH refers to a "lithium magnesium silicate solution." It was defined in section 3, making it difficult for the reader to interpret. It should be defined once and at the beginning of the text.
Answer 2: Thank you for your question. The LMSH has been defined once and at the beginning of the text.
Question 3: The text failed to define CK.
Answer 3: Thank you for your question. CK was redefined in this text.
Question 4: Figure 2 ddH2O+Lf+NZ2114 is not defined in the text.
Answer 4: Thank you for your question. The “ddH2O+Lf+NZ2114” is “Lf+NZ2114”, it was changed to Lf+NZ2114 in Figure 2.
Question 5: The text requires careful revision to improve the organization of ideas and address grammatical issues.
Answer 5: Thank you for your question. The text has been revised to improve the organization of ideas and address grammatical issues.
Question 6: There are instances when the caption and lettering in the figure do not align. For instance, Figure 1b and its caption correspond with the description label as c.
Answer 6: Thank you for your question. The errors have been revised in Figure 1.
Comments on the Quality of English Language
The text requires careful revision to improve the organization of ideas and address grammatical issues.
Answer: The text has been revised to improve the organization of ideas and address grammatical issues.

Reviewer 3 Report
Comments and Suggestions for Authors
The objective of this study was the preparation of bifunctional injectable hydrogel Lf/NZ2114/LMSH wound dressing (antimicrobial, anti-inflammatory) for promoting Staphylococcus aureus infected wound healing. The research described is organized and the results obtained are relevant. Even though this manuscript has critical points that need to be clarified by the authors. These points are outlined below:
1- The abbreviation for the hydrogel should be defined in the abstract like LMSH. In the main text as well sometimes the authors used abbreviations or full names, you need to keep the same context to facilitate for the readers.
2- How do you provide a solution for traditional hydrogel to fix the hydrogel on the wound location? What are the expected mucoadhesive properties of the prepared hydrogel?
3- What was the temperature of the Rheological analysis?
4- The preparation of hydrogel is not clear, we have different compositions with different concentrations (not equal; 3 conc for LMSH, 5 conc for Lf, 4 conc for NZ2114) and not clearly stating how the final product is prepared, I suggest using the table with the used composition! What is the main charge of the formed hydrogel, and how it will be affected by the pH changes?
5- Line 141, amide I have a higher wavenumber compared to amide II, not as mentioned by the authors, please revise!
6- Figure 1, the caption order is not the same as the order of the figure and main text, please revise!
The presentation of FITR is useless in its current form, please highlight the interesting peaks that confirm the composition of the hydrogel and remove the legend because you already wrote the names on the curves and increase the size of Figure 1C to be able to distinguish the difference in the peaks.
7- Lines 169-171, “The results of rheological characterization of the 1 % Lf/NZ2114/LMSH and 1.5 % Lf/NZ2114/LMSH groups are shown in the following table.” Where is the table?
8- Figure 2c, what is the reason for increasing G” for 3%HACC compared to G’, and G” shows the stability with an increase in the strain but G’ not? G” and G’ can be presented in the same axis better than the current form. The injectability test should be presented in Figure 2, what type of syringe should be used for your hydrogel (single or dual)? What is the gelation time for the prepared hydrogels?
9- Line 325, what is the meaning of the “In conclusion, the prepared water for injection gel”, you need to revise it!
10- To get the full picture, this system can be applied to wound healing. What are the application scenarios for this system? Does it need to be used once or should it be applied to wounds repeatedly? Does it need surgical tape to fix the hydrogel or not?
11- What about the in vitro/in vivo drug release for your system?
Comments on the Quality of English LanguageQuality of English language is fine, but can be improved as well!
Author Response
Reviewer 3
Comments and Suggestions for Authors
The objective of this study was the preparation of bifunctional injectable hydrogel Lf/NZ2114/LMSH wound dressing (antimicrobial, anti-inflammatory) for promoting Staphylococcus aureus infected wound healing. The research described is organized and the results obtained are relevant. Even though this manuscript has critical points that need to be clarified by the authors. These points are outlined below:
Question 1: The abbreviation for the hydrogel should be defined in the abstract like LMSH. In the main text as well sometimes the authors used abbreviations or full names, you need to keep the same context to facilitate for the readers.
Answer 1: Thank you for your question. All abbreviations have been revised and kept the same context.
Question 2: How do you provide a solution for traditional hydrogel to fix the hydrogel on the wound location? What are the expected mucoadhesive properties of the prepared hydrogel?
Answer 2: Thank you for your question. By experimenting with different LF, AMP and LMSH concentrations and ratios, the optimal ratio of hydrogel consistency was chosen to allow for good adhesion to the wound location. Finally, the prepared injectable hydrogel 1% Lf/NZ2114/LMSH possesses shear-thinning characteristics, which can be injected and gelated in vivo to exert antimicrobial and pro-healing effects, avoiding the drawbacks of the traditional hydrogel, such as low adhesion and easy to fall off.
Question 3: What was the temperature of the Rheological analysis?
Answer 3: Thank you for your question. The temperature of the Rheological analysis was room temperature.
Question 4: The preparation of hydrogel is not clear, we have different compositions with different concentrations (not equal; 3 conc for LMSH, 5 conc for Lf, 4 conc for NZ2114) and not clearly stating how the final product is prepared, I suggest using the table with the used composition! What is the main charge of the formed hydrogel, and how it will be affected by the pH changes?
Answer 4: Thank you for your question. The proportions of hydrogel were described in detail: “Lf and NZ2114 were put in a 100 mL beaker, The appropriate ddH2O was added with final concentration were 20 mg/mL, 40 mg/mL, 80 mg/mL, 160 mg/mL, 320 mg/mL and 200 µg/mL, 400 µg/mL, 800 µg /mL1600 µg/mL respectively, stirred in a magnetic ag-itator until fully dissolved, and the pH value of the solution was adjusted to 6–7. LMSH was placed in a 50 mL beaker, The appropriate ddH2O was added, stirred in a magnetic agitator for 30 min, and completely dissolved by ultrasonication at room temperature for another 30 min. The Lf/NZ2114 was mixed with LMSH, and the final concentration of LMSH was 0.5%, 1%, 1.5%. 0.5%, 1%, 1.5% lithium magnesium silicate (LMSH) was con-figured and mixed with final concentrations of 20 mg/mL, 40 mg/mL, 80 mg/mL, 160 mg/mL, 320 mg/mL lactoferrin (Lf) respectively by thorough stirring and ultrasonication to homogeneity; the same as, with final concentrations of 200 µg/mL, 400 µg/mL, 800 µg /mL1600 µg/mL NZ2114 were thoroughly stirred and sonicated to homogeneity. Different combinations of gel-forming states were observed, and the lowest gel-forming concentra-tions of lithium magnesium silicate, lactoferrin, and NZ2114 were screened. Finally, it was found that 1.5%, 1% LMSH were ultrasonically mixed with 40 mg/mL Lf and 400 µg/mL NZ2114 to prepare Lf/NZ2114/LMSH hydrogels for follow-up studies.”
Question 5: Line 141, amide I have a higher wavenumber compared to amide II, not as mentioned by the authors, please revise!
Answer 5: Thank you for your question. The amide group I had characteristic absorption peaks at 1550 nm, and it is described in the text.
Question 6: Figure 1, the caption order is not the same as the order of the figure and main text, please revise!
The presentation of FITR is useless in its current form, please highlight the interesting peaks that confirm the composition of the hydrogel and remove the legend because you already wrote the names on the curves and increase the size of Figure 1C to be able to distinguish the difference in the peaks.
Answer 6: Thank you for your question. The caption order was revised according to figure 1. The results of the FT-IR test showed that the characteristic spectral bands of the materials forming the injectable hydrogels did not change before and after gel formation, indicating that the binding mode was physical cross-linking: electrostatic interactions. The legend was removed in figure 1.
Question 7: Lines 169-171, “The results of rheological characterization of the 1 % Lf/NZ2114/LMSH and 1.5 % Lf/NZ2114/LMSH groups are shown in the following table.” Where is the table?
Answer 7: Thank you for your question. The lines 169-171 were rephrased, we made a writing mistake, the table was changed to Fig. 3c.
Question 8: Figure 2c, what is the reason for increasing G” for 3%HACC compared to G’, and G” shows the stability with an increase in the strain but G’ not? G” and G’ can be presented in the same axis better than the current form. The injectability test should be presented in Figure 2, what type of syringe should be used for your hydrogel (single or dual)? What is the gelation time for the prepared hydrogels?
Answer 8: Thank you for your question. The storage modulus G‘ of 3% Lf/NZ2114/HACC is smaller than the release modulus G’, and the gel tends to be in the liquid state, and the storage modulus G‘ and release modulus G’ basically remain unchanged with the increase of shear, indicating that it does not have the characteristics of shear thinning. While the storage modulus (G′) of the 1%/Lf/LMSH group decreased with increasing shear strain, while its loss modulus (G″) first increased with increasing shear strain, undergoing a sol-gel transition at approximately 10 % of shear strain, from storage modulus G′ < loss modulus G″, after which the storage modulus G″ and loss modulus G″ decreased sharply, indicating significant shear thinning properties. These shear thinning properties allow the hydrogel to be easily injected into the target site through a fine needle. Therefore, 1% Lf/NZ2114/LMSH injectable hydrogel was selected. The gelation time was within 5 s for the prepared hydrogels.
Question 9: Line 325, what is the meaning of the “In conclusion, the prepared water for injection gel”, you need to revise it!
Answer 9: Thank you for your question. The “In conclusion, the prepared water for injection gel” was revised to “In conclusion, the injectable hydrogel”
Question 10: To get the full picture, this system can be applied to wound healing. What are the application scenarios for this system? Does it need to be used once or should it be applied to wounds repeatedly? Does it need surgical tape to fix the hydrogel or not?
Answer 10: Thank you for your question. This hydrogel can be used for wounds with varying degrees of infection, applying or injecting this hydrogel depending on the degree of infection in the wound, and can be reapplied many times until complete healing is achieved, this injectable hydrogel does not require a surgical tape to fix.
Question 11: What about the in vitro/in vivo drug release for your system?
Answer 11: Thank you for your question. In this paper, the drug release was verified in Fig. 1 Inhibition zone, but the hydrogel composition is complex and not easy to be detected, in this paper, we only verify that the hydrogel can be released well by the results of antimicrobial activity and pro-healing test, and we may further establish the method to determine the release at a later stage.
Comments on the Quality of English Language
Quality of English language is fine, but can be improved as well!
Answer: Quality of English language has been improved possibility.

Reviewer 4 Report
Comments and Suggestions for Authors
In this work, Zhang et al. designed a bifunctional injectable hydrogel incorporating lactoferrin (Lf), NZ2114, and LMSH, which demonstrated strong antimicrobial efficacy and biocompatibility, both in vitro and in vivo. While the study is comprehensive and merits consideration for publication, the following points need to be addressed:
1. The sequence of NZ2114 should be included.
2. Figure 1a is unclear. If the figure represents four types of hydrogels (0.25% LMSH, 0.5% LMSH, 1% LMSH, and 1.5% LMSH), please clearly label them.
3. In Figure 1, the captions for Figures 1b and 1c appear to be reversed.
4. In Figure 2, the scale bars are difficult to discern.
5. For Figure 3d, it is recommended to use distinct labels for the samples instead of written annotations, as the sample names are hard to distinguish.
6. In Figure 3d, it is challenging to identify individual colonies. How was the antimicrobial rate in Figure 3e calculated?
7. There is no statistical significance indicated in Figures 4b and 4c. Please check for statistical significance across all figures.
Comments on the Quality of English LanguageIn this work, Zhang et al. designed a bifunctional injectable hydrogel incorporating lactoferrin (Lf), NZ2114, and LMSH, which demonstrated strong antimicrobial efficacy and biocompatibility, both in vitro and in vivo. While the study is comprehensive and merits consideration for publication, the following points need to be addressed:
1. The sequence of NZ2114 should be included.
2. Figure 1a is unclear. If the figure represents four types of hydrogels (0.25% LMSH, 0.5% LMSH, 1% LMSH, and 1.5% LMSH), please clearly label them.
3. In Figure 1, the captions for Figures 1b and 1c appear to be reversed.
4. In Figure 2, the scale bars are difficult to discern.
5. For Figure 3d, it is recommended to use distinct labels for the samples instead of written annotations, as the sample names are hard to distinguish.
6. In Figure 3d, it is challenging to identify individual colonies. How was the antimicrobial rate in Figure 3e calculated?
7. There is no statistical significance indicated in Figures 4b and 4c. Please check for statistical significance across all figures.
Author Response
Reviewer 4
Comments and Suggestions for Authors
In this work, Zhang et al. designed a bifunctional injectable hydrogel incorporating lactoferrin (Lf), NZ2114, and LMSH, which demonstrated strong antimicrobial efficacy and biocompatibility, both in vitro and in vivo. While the study is comprehensive and merits consideration for publication, the following points need to be addressed:
Question 1: The sequence of NZ2114 should be included.
Answer 1: Thank you for your question. The sequence of NZ2114 was pdb number: 6K50 “GFGCNGPWNEDDLR CHNHCKSIKGYKGGYCAKGGFVCKCY”.
Question 2: Figure 1a is unclear. If the figure represents four types of hydrogels (0.25% LMSH, 0.5% LMSH, 1% LMSH, and 1.5% LMSH), please clearly label them.
Answer 2: Thank you for your question. The label has been added.
Question 3: In Figure 1, the captions for Figures 1b and 1c appear to be reversed.
Answer 3: Thank you for your question. The captions of Figure 1 have been revised.
Question 4: In Figure 2, the scale bars are difficult to discern.
Answer 4: Thank you for your question. Each picture has a scale, the scale may not be clearly observed due to the large number of pictures displayed, but the pictures clearly reflect the structure and are under a uniform scale.
Question 5: For Figure 3d, it is recommended to use distinct labels for the samples instead of written annotations, as the sample names are hard to distinguish.
Answer 5: Thank you for your question. The Figure 3d has been revised.
Question 6: In Figure 3d, it is challenging to identify individual colonies. How was the antimicrobial rate in Figure 3e calculated?
Answer 6: Thank you for your question. We calculated antimicrobial rate based on the Figure 3d colonies.
Question 7: There is no statistical significance indicated in Figures 4b and 4c. Please check for statistical significance across all figures.
Answer 7: Thank you for your question. The statistical significance has been added.

Round 2
Reviewer 1 Report
Comments and Suggestions for Authors
Authors have improved the manuscript considerably and i recommend publication of the research.
Comments on the Quality of English LanguageTypological errors and repetative sentences were improved, however still some changes can be done for furhter refinining the sentences.
Reviewer 2 Report
Comments and Suggestions for Authors
Dear Editor,
I believe that the revisions made by the authors have markedly enhanced the quality and clarity of the manuscript. Therefore, I recommend that it be accepted for publication in Antibiotics.